# Identification of TRPC6 as a Novel Diagnostic Biomarker of PM-Induced Chronic Obstructive Pulmonary Disease Using Machine Learning Models

**DOI:** 10.3390/genes14020284

**Published:** 2023-01-21

**Authors:** Kyu-Ree Dhong, Jae-Hyeong Lee, You-Rim Yoon, Hye-Jin Park

**Affiliations:** 1Department of Food Science and Biotechnology, College of BioNano Technology, Gachon University, 1342 Seongnam-daero, Sujeong-gu, Seongnam-si 13120, Republic of Korea; 2Department of Computer Engineering, College of Information Technology, Gachon University, 1342 Seongnam-daero, Sujeong-gu, Seongnam-si 13120, Republic of Korea

**Keywords:** TRPC6, particulate matter, chronic obstructive pulmonary disease (COPD), machine learning

## Abstract

Chronic obstructive pulmonary disease (COPD) was the third most prevalent cause of mortality worldwide in 2010; it results from a progressive and fatal deterioration of lung function because of cigarette smoking and particulate matter (PM). Therefore, it is important to identify molecular biomarkers that can diagnose the COPD phenotype to plan therapeutic efficacy. To identify potential novel biomarkers of COPD, we first obtained COPD and the normal lung tissue gene expression dataset GSE151052 from the NCBI Gene Expression Omnibus (GEO). A total of 250 differentially expressed genes (DEGs) were investigated and analyzed using GEO2R, gene ontology (GO) functional annotation, and Kyoto Encyclopedia of Genes and Genomes (KEGG) identification. The GEO2R analysis revealed that TRPC6 was the sixth most highly expressed gene in patients with COPD. The GO analysis indicated that the upregulated DEGs were mainly concentrated in the plasma membrane, transcription, and DNA binding. The KEGG pathway analysis indicated that the upregulated DEGs were mainly involved in pathways related to cancer and axon guidance. TRPC6, one of the most abundant genes among the top 10 differentially expressed total RNAs (fold change ≥ 1.5) between the COPD and normal groups, was selected as a novel COPD biomarker based on the results of the GEO dataset and analysis using machine learning models. The upregulation of TRPC6 was verified in PM-stimulated RAW264.7 cells, which mimicked COPD conditions, compared to untreated RAW264.7 cells by a quantitative reverse transcription polymerase chain reaction. In conclusion, our study suggests that TRPC6 can be regarded as a potential novel biomarker for COPD pathogenesis.

## 1. Introduction

Chronic obstructive pulmonary disease (COPD) is a major lung disease and the third leading cause of death worldwide in 2010 [1]. It is an abnormal inflammatory response caused by exposure to particulate matter (PM), including toxic particles and gases. Patients with COPD experience chronic cough, chronic bronchitis, and accelerated lung dysfunction. Many studies have reported that a significant increase in inflammatory immune cells has been observed in the small airways of patients with COPD, which release fatal enzymes and inflammatory factors, leading to lung damage [2]. Neutrophils play major roles in COPD by producing neutrophil elastase (NE) and myeloperoxidase (MPO) [3]. Macrophages exacerbate COPD by releasing excess proinflammatory cytokines, matrix metalloproteinases (MMP), and reactive oxygen species (ROS) [4].

Examining the sputum and bronchoalveolar lavage fluid (BALF) of COPD patients for clinical applications is difficult. Hence, the detection of serum inflammatory markers in COPD patients is mainly used in clinical practice and correlates inflammatory markers with the severity of COPD. Thomsen et al. found that C-reactive protein (CRP), fibrinogen, and leukocyte population could be important COPD biomarkers associated with increased exacerbation [5]. However, the study had a reliability issue in the experimental results because of the unbalanced and too few non-viable samples.

To clearly distinguish the severity of COPD, many scientists have attempted to use machine learning algorithms for clinical decision making. Nunavath et al. investigated feed-forward neural networks (FFNN) for COPD classification and long short-term memory (LSTM) for the early prediction of exacerbation degrees and subsequent triage in patients with COPD [6]. However, the data were obtained from a family environment, which was likely to be affected by multiple factors, resulting in diminishing data quality. Tang et al. suggested a four-layer deep learning model that optimizes a specifically configured recurrent neural network to address temporal variations in COPD progression [7]. The proposed model led to a poor interpretation owing to its complexity. Almagro et al. utilized the Charlson index and a questionnaire to investigate comorbidities and short-term prognosis in hospitalized patients with COPD with exacerbation [8]. That study did not include the role of inflammation in the diagnosis of COPD.

In this study, we aimed to identify novel potential diagnostic biomarkers of COPD through machine learning models a database analysis of the largest publicly available repository of mRNA expression in COPD collected by the Gene Expression Omnibus (GEO) (https://www.ncbi.nlm.nih.gov/geo/ accessed on 10 January 2022) and revealed their expression in macrophages with PM-induced COPD (Figure 1). 

## 2. Materials and Methods

### 2.1. Microarray Data Acquisition

GEO (http://www.ncbi.nlm.nih.gov/geo accessed on 10 January 2022) provides genomics data including high throughout microarrays and gene expression data to the public. One gene expression dataset [GSE151052] was used from GEO (GPL17556 [HuGene-1_0-st] Affymetrix Human Genome 1.0 ST Array). According to the annotation information in the platform, the probes are transformed into corresponding gene symbols. The GSE151052 dataset contained the total RNAs of 117 samples, including 78 samples from lung tissues of COPD patients and 39 samples from control.

### 2.2. Identification of Differentially Expressed Genes (DEGs)

GEO2R (http://www.ncbi.nlm.nih.gov/geo/geo2r/ accessed on 10 January 2022) is a web tool based on the R language limma package to obtain differentially expressed genes (DEGs) for comparing more than two groups of samples [9]. We utilized this tool to conduct comparisons on GSE151052 raw data. For this, we initially checked the overall characteristics of value distributions. Usually, median-centered values mean that the data are normalized. If the data were not normalized, the force normalization option was applied for quantile normalization to the expression data, forcing all selected samples to show identical value distribution. Then, we assigned samples from COPD patients and normal to “case group” and “control group”, respectively. Differentially expressed genes (DEGs) were identified using GEO2R (http://www.ncbi.nlm.nih.gov/geo/geo2r/ accessed on 10 January 2022), which is based on the GEO databases. In order to classify the DEGs among patients with COPD and controls, DEGs were acquired by |log (fold change; FC) | >1 and t-tests with *p* < 0.05.

### 2.3. GO Enrichment and KEGG Pathway Analysis

The Database for Annotation, Visualization, and Integrated Discovery (DAVID) v6.8 tool was used to interpret the functional roles of genes based on the genome studies [10]. The Gene Ontology database (GO; http://www.geneontology.org accessed on 15 February 2022) provides structured ontologies or vocabularies, explaining the characteristics of genes and gene products [11]. Kyoto Encyclopedia of Genes and Genomes database (KEGG; (http://www.genome.jp/kegg/ accessed on 15 February 2022)) provides information on biological systems from genomic, systemic functional, and chemical points [12]. We analyzed the GSE151052 database using DAVID software, according to Han’s report [13]. In our study, in the first step, we input the gene list into the search box, subsequently selected identifier “ENSEMBL_GENE_ID” and chose list type “Gene List”, then submitted the list. In the second step, we selected “Homo sapiens” to limit annotations and selected “List 1”. In the third step, we chose the four parameters “GOTERM-BP-DIRECT”, “GOTERM-CC-DIRECT”, “GOTERM-MF-DIRECT”, and “KEGG-PATHWAY” in the annotation summary results. To determine GO an KEGG pathway analysis results, “Functional Annotation Chart” was used.

### 2.4. Cell Culture

The RAW 264.7 cells were obtained from the Korean Cell Line Bank (KCLB, Seoul, Republic of Korea). Cells were cultured in Dulbecco’s modified Eagle’s medium (DMEM, Welgene, Daegu, Republic of Korea) supplemented with 10% fetal bovine serum (Welgene) and 1% penicillin and streptomycin (Welgene). They were grown in a 75 cm^2^ cell culture flask at 37.5 °C in humidified 5% CO_2_ incubators.

### 2.5. Extraction of RNAs and qRT-PCR

According to a previous study [14,15], total mRNA was extracted from RAW 264.7 cells using TRIzol reagent (Invitrogen, Carlsbad, CA, USA) and reverse transcribed using Revertra Ace qPCR RT kit (Toyobo Biologics Inc., Osaka, Japan). The PCR program was implemented as follows: initial denaturation at 94 °C for 2 min, followed by 30 cycles of denaturation at 94 °C for 20 s, annealing at 62.2 °C for 10 s, and extension at 72 °C for 45 s, with a final extension at 72 °C for 5 min. Polymerase chain reaction was performed at 94 °C for 2 min, 94 °C for 30 s, 55 °C for 30 s, and 68 °C for 1 min for 30 cycles. The sequence of TRPC6: forward 5′- GAA CTT AGC AAT GAG CTG GC -3′ and reverse5′- CAG AGG TCC AAG AGA CCA AC -3′. The levels of TRPC6 mRNA were normalized to GAPDH mRNA.

### 2.6. Statistical Analysis

The data were expressed as means ± standard deviation (SD) and analyzed by one-way ANOVA/Duncan’s t-test. These analyses were performed using the SPSS software program, version 12 (SPSS Inc., Chicago, IL, USA).

### 2.7. Machine Learning with Decision Tree

Machine learning is a technique to optimize compute performance criteria that use data or experience [16]. It provides possible solutions to detect the information hidden in massive and complex data [17]. Applying machine learning is an appropriate way to analyze microarray data, which are expressions of measurements of thousands of genes, and select the necessary genes from the microarray [17,18].

However, many machine learning models lack the explanatory ability for results, but decision tree models make it easy to identify the criteria for classification problems [14]. Therefore, we used decision tree [15] algorithms in this study. Decision tree algorithms allow the identification of the criteria for classifying COPD efficiently. When applying a decision tree algorithm with datasets, the algorithm generates tree-based classification criteria [18]. In other words, when we apply this algorithm with microarray data, the algorithm creates a tree-based COPD classification criterion based on the gene expression amount contained in the data. It makes it possible to visually check which genes greatly influence COPD classification and how much gene expression is the criterion for disease classification. Figure 2 shows a brief schema of decision tree.

The decision tree algorithm can be also used to classify microarray data because it outputs easy-to-understand results without generating complex rules that are difficult to analyze from a medical point of view. It does not require a complicated parameter-tuning process [19,20]. It is also a suitable method for biological data analysis in that the results obtained through this method can be treated as valuable information for further analysis [21]. Because of this feature, decision tree-based algorithms can be used as effective methodologies for microarray-based data analysis, such as those used for direct disease classification [19,21] or target gene screening [20,22].

Among them, the J48 algorithm, a representative decision tree algorithm, is an implementation of the C4.5 algorithm (revision 8) by Ross Quinlan [23]. The C4.5 algorithm is a decision tree algorithm and an improvement of the ID3 algorithm. Like ID3, C4.5 also uses formulas based on information theory and evaluates the goodness of a test with them, under the criterion of selecting a test that can extract maximum extractable information from a set of cases, considering constraints in which only one attribute is tested [24]. C4.5 shows several improvements over ID3, such as continuous data and unknown values that can be used for the algorithm’s input and attributes with different weights. Furthermore, due to pruning that is carried out after creation of the tree, the algorithm is enabled for pessimistic error prediction, subtree raising to simplify the tree by delete node, replacing it with the sub-tree, and redistributing instances with its classification criteria [25].

C4.5 is a form of a greedy technique that is a top-down recursive divide-and-conquer form of approach [26]. The algorithm selects specific well-classified values, separates them into child nodes, and recursively invokes the algorithm per sub-node basis [23]. Figure 3 shows its pseudocode.

## 3. Results

### 3.1. Identification of TRPC6 as a Potential Biomarker for COPD Using Machine Learning Models and GEO2R

#### 3.1.1. Analysis Using Machine Learning Models

We conducted machine learning to identify a classifier capable of identifying COPD from the microarray data, and to identify important genes for COPD classification. The GSE151052 dataset used in this study comprises microarray profiles of 77 COPD samples and 40 control samples extracted from the lungs of patients with COPD and control group donors. Each sample contained information on 19,718 DEGs. A decision tree, which is a type of a machine learning model, can be used to classify microarray profiling data into the two groups (COPD and control).

In this study, we generated decision tree models for classifying COPD gene expression data and investigated the genes that were crucial to classification by analyzing the generated decision tree structures. We used the J48, DecisionStump, and REPTree models [27] implemented in WEKA [28]. Owing to the small amount of data, the results were verified using 10-fold cross validation [23]. We evaluated each classifier using the accuracy and F1 scores that were derived based on a confusion matrix [29], as shown in Table 1.

The Accuracy (Acc) and F1 Scores (F_1) are calculated as follows:$$Acc=\frac{TP+TN}{TP+FP+TN+FN}\;F_{1}=\frac{2PR}{P+R}\;P=\frac{TP}{TP+FP}\;R=\frac{TP}{TP+FN}$$

Figure 4 and Figure 5 show the tree structures obtained by learning the three decision tree models and their performances—J48, DecisionStump, and REPTree; all three models classified the data using only the value of gene ID 7225_at. These figures explain that algorithms that specially concentrated in 7225_at among genes in the microarray data while making decision trees classifying the disease. Moreover, these classifiers, using only 7225_at, showed significant performance in the COPD microarray profile dataset, with an accuracy of up to 0.991.

Originally, a decision tree is a machine learning model used to classify groups, and it is impossible to calculate the validation rate of each gene or rank the genes using the model. However, it is possible to infer the importance of genes in classification by analyzing the structure of the tree optimized for classification. Figure 4 shows that all three decision tree models optimized for the classification of COPD contained only one TRPC6 gene (7225_at). The result is unusual because machine learning results using a conventional microarray or RNA-sequencing generally do not determine disease classification using only one gene. Figure 6 and Figure 7 show the results of the same experiment but with the GSE57148 [30] data, and revealed that the results with the J48 classifier came in a complex tree form while at the same time performing poorly overall. The results also indicated that all three classifiers were focused on different genes. The results also differed from those of the original experiment in which most classifiers focused on only one gene. Therefore, Figure d suggests that there is something to pay attention to in the results from the original experiment.

The statistical analysis results (Table 1) were ranked using the statistical value of LogFC, and the ranking was not absolute. If the results were re-ranked with *p*-value, another result would be obtained. However, according to the results of the decision trees, there was strong evidence that COPD can be classified using only one TRPC6 gene. All three decision tree models with dataset GSE151052 included only one TRPC6 gene (Figure 5), and their classification accuracies were very high, at over 98%. Hence, the results of the decision trees were considered to have priority over those of statistical analysis in this study.

The results of the decision trees with dataset GSE151052 showed that COPD could be classified with only TRPC6 alone regardless of other genes, so we appraised TRPC6 as a definite biomarker. In practice, it is very rare that samples are classified with high accuracy based on only one gene in a decision tree. The results with the J48 decision tree using another dataset, GSE57148 (Figure 5), which does not contain TRPC6, showed that many genes were included in the decision tree. However, all three decision tree models with dataset GSE151052 included only one TRPC6 gene (Figure 5), and their classification accuracies were very high, at over 98%. Since this is strong evidence that COPD can be classified using only one TRPC6 gene, the results of the decision trees were considered to have priority over those of statistical analysis.

Typically, a decision tree algorithm applies pruning to deal with overfitting in the machine learning process. The algorithm produces an optimal decision tree through pruning, which soon shows which gene the algorithm is paying attention to [30]. According to the presented experimental results, all three algorithms produced classifiers by focusing on the 7225_at, which is termed TRPC6, and the accuracy of each classifier was also relatively high. This result shows that the algorithms generated the correct classifiers with only the TRPC6. Moreover, the experimental results suggest that TRPC6 plays a crucial role in terms of COPD classification using machine learning methods. This finding implies that TRPC6 is very important in terms of classification using machine learning and further suggests that TRPC6 can also be considered as a biomarker of COPD pathogenesis.

#### 3.1.2. GEO2R

After the analysis of differentially expressed RNA in the GSE151052 (*n* = 117) dataset, a total of 250 DEGs were identified, of which 15 genes were upregulated and 12 genes were downregulated in patients with COPD compared to the normal group (Table 2). The machine learning results indicated that TRPC6 was the most highly expressed gene among the top 10 upregulated genes in patients with COPD compared to the control. The GEO2R analysis showed that RTKN2 was the most abundant among the top 10 upregulated genes, and TRPC6 was the eighth most abundant gene among the top 10 upregulated genes in patients with COPD compared to the control.

### 3.2. GO Term Enrichment and KEGG Pathway Analysis

To identify significant functional DEGs between the COPD and normal groups, DAVID software was used. The top five enrichment analyses for GO are shown in Table 3 and Table 4. For the biological process (BP) enrichment analysis, the upregulated genes were significantly involved in the transcription: DNA-templated (GO:0006351), transmembrane transport (GO:0055085), post-embryonic development (GO:0009791), ossification (GO:0001503), and covalent chromatin modification (GO:0016569). In addition, the TRPC6 gene was included in the manganese ion transport (GO:0006828).

For the cell component (CC) enrichment analysis, upregulated genes in patients with COPD were enriched in the plasma membrane (GO:0005886), intracellular (GO:0005622), nuclear chromatin (GO:0000785), sarcolemma (GO:0042383), and receptor complex (GO:0043235). TRPC6 was localized in the plasma membrane (GO:0005886).

For the molecular function (MF), upregulated genes were mainly involved in the DNA binding (GO:0003677), transcription factor activity, sequence-specific DNA binding (GO:0003700), calcium ion binding (GO:0005509), chromatin binding (GO:0003682), and integrin binding (GO:0005178). The TRPC6 gene was involved in inositol 1,4,5 trisphosphate binding (GO:0070679) and store-operated calcium channel activity (GO:0015279).

The KEGG pathway analysis showed that the upregulated DEGs were mainly enriched in axon guidance (map04360), serotonergic synapse (map04726), and pathways in cancer (map05200). The downregulated DEGs were mainly enriched in the biosynthesis of antibiotics (map00998), metabolic pathways (map01100), biosynthesis of amino acids (map01230), carbon metabolism (map01200), and aminoacyl-tRNA biosynthesis(map00970). However, TRPC6, which is upregulated in patients with COPD, was not included in the three pathways of upregulated DEGs.

### 3.3. Validating the Expression and Diagnostic Value of TRPC6 in Vitro Model of COPD Using PCR Analysis

Recently, many groups have reported that particulate air pollution, including PM10, is a major risk factor for COPD [31]. It stimulates immune cells in the lungs, such as alveolar macrophages [22]. To confirm TRPC6, which was identified as a novel biomarker of COPD based on the GEO2R analysis and machine learning analysis, we investigated the level of TRPC6 mRNA expression in PM-stimulated RAW 264.7 macrophages. The level of TRPC6 mRNA expression was significantly upregulated in the PM-stimulated RAW 264.7 macrophages compared to the control (*p* < 0.05) (Figure 8).

## 4. Discussion

COPD is a major disease with a steep increase in morbidity and mortality rates worldwide. According to COPD studies, the levels of IL-6, IL-8, tumor necrosis factor (TNF)-α, CRP, fibrinogen, and leukocyte population could be considered COPD biomarkers [12,32]. Many studies have implied that a large surge in COPD incidence is due to air pollution, such as PM. In vitro and in vivo studies have shown that PM can induce pulmonary inflammation, destroy lung function, cause emphysematous changes in PM10, and induce the release of proinflammatory cytokines (e.g., TNF-α and IL-1) and reactive oxygen radicals by alveolar macrophages in patients with COPD [31]. These biomarkers of COPD, based on the literature review, have been studied for decades. However, they cannot reflect with certainty the severity of COPD pathophysiology in current clinical practice. To identify new biomarkers in PM-induced COPD, we used GEO2R and machine learning methods. In our study, the gene expression data of GSE6676 were downloaded to identify novel biomarkers that were differentially expressed in the lungs of patients with COPD versus healthy people. Our study indicated that there were TRPC6 DEGs between the COPD and normal groups.

In this study, an unusual result was obtained from the analysis of COPD microarray data using decision tree machine learning models. A decision tree learned for a classification problem usually has many nodes and a depth of three or more. However, in this case, a simple binary tree structure with only three nodes and a depth of two was obtained, in which the COPD and control groups were classified with more than 99% accuracy. In this decision tree, only one gene (TRPC6) was used for the classification. Hence, the effect of TRPC6 was significantly greater in the classification of COPD than that of other genes. This finding contradicts the results of the analysis using GEO2R. In the analysis using GEO2R, TRPC6 was ranked sixth. In addition, TRPC6 has not been identified as a major biomarker for COPD in previous studies. Thus, new biomarker candidates that cannot be found using statistical methods such as GEO2R can be identified by some machine learning methods.

To investigate the predicted biological functions and signaling pathways of the DEGs in patients with COPD, we performed GO and KEGG pathway analyses. The GO analysis indicated that upregulated DEGs mainly participated in the BP and MF, whereas downregulated DEGs mainly took part in the BP. The GO analysis showed that the predicted targets of COPD were mainly enriched for transcription (DNA-templated), plasma membrane, and DNA binding. TRPC6 is localized in the plasma membranes.

TRPC6, which is a Ca^2+^-permeable cation and an oxidative stress-sensitive channel located in the plasma membrane, is widely expressed in various tissues. TRPC-dependent increases in Ca2+ in pulmonary cells induce the activation of inflammatory signaling molecules (e.g., ERK1/2, p38, and JNK), which increases the levels of the inflammatory factors IL-6 and IL-8 in COPD [33]. TRPC6 is expressed in the lungs, including bronchial epithelial cells, alveolar macrophages, and the pulmonary vasculature [34]. Finney-Hayward et al. reported that the level of TRPC6 mRNA in alveolar macrophages from patients is significantly higher than that in healthy controls [35]. Therefore, TRPC6 identified from GEO2R and machine learning analysis could be a novel biomarker for the pathogenesis of COPD.

Studies have demonstrated that macrophages are the major cell type in COPD [36]. Macrophages are innate effector cells for pulmonary host defense against pathogeneses and inhaled particles such as PM. The number of macrophages was significantly increased (5- to 10-fold) in the airways, bronchial tubes, and BALF of patients with COPD [37,38]. In addition, a positive correlation was shown between the number of macrophages in the airways and COPD severity [39]. Recently, many groups have reported that particulate air pollution, including PM10, is a major risk factor for COPD by stimulating alveolar macrophages [40]. We found that TRPC6 is a significantly increased molecule in patients with COPD using machine learning methods and GEO2R. To determine whether TRPC6 is upregulated in COPD, we investigated the level of TRPC6 mRNA expression in RAW 264.7 macrophage-stimulated PM and analogical COPD condition. The experimental verification showed that the level of TRPC6 mRNA expression in the PM-stimulated RAW 264.7 cells was increased in a concentration-dependent manner. This result suggests that TRPC6 is significantly expressed in the pathogenesis of COPD.

## 5. Conclusions

Our study suggests that TRPC6 can be regarded as a potential novel biomarker for COPD pathogenesis. All three machine learning algorithms (J48, DecisionStump and REPTree) suggested that TRPC6 plays a crucial role in terms of COPD classification. The mRNA expression of TRPC6 is significantly increased in PM-stimulated RAW264.7 cells, which mimic COPD. For diseases other than COPD, a method for deriving biomarker candidates using machine learning and microarray data can be effective. Research on diverse gene expression data is left for future works.

## Figures and Tables

**Figure 1 genes-14-00284-f001:**
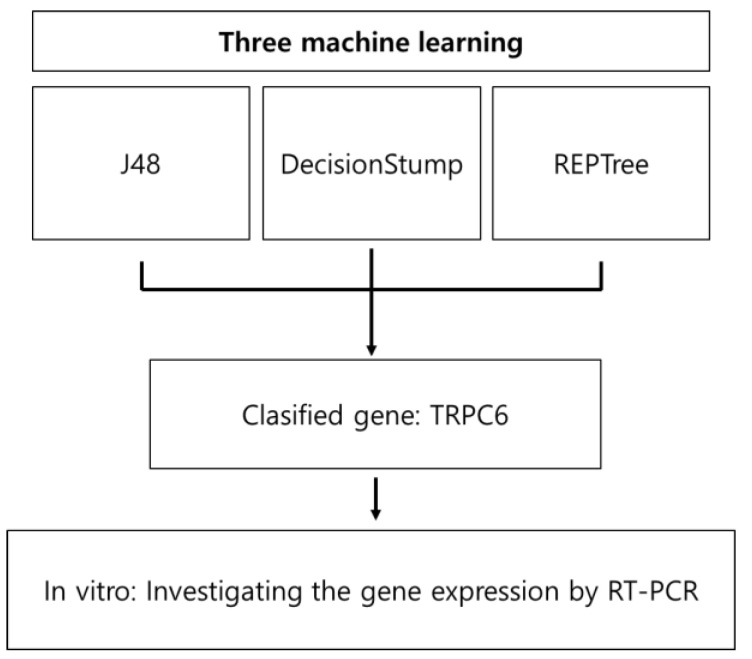
Overview of the experimental workflow.

**Figure 2 genes-14-00284-f002:**
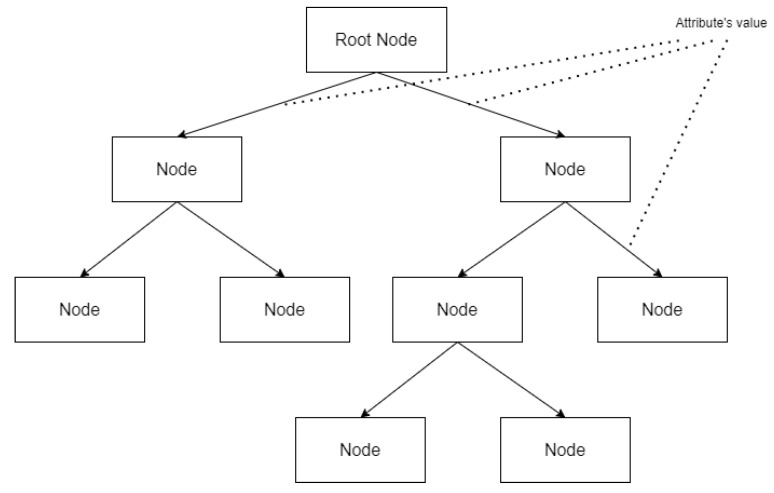
A brief schema of decision tree.

**Figure 3 genes-14-00284-f003:**
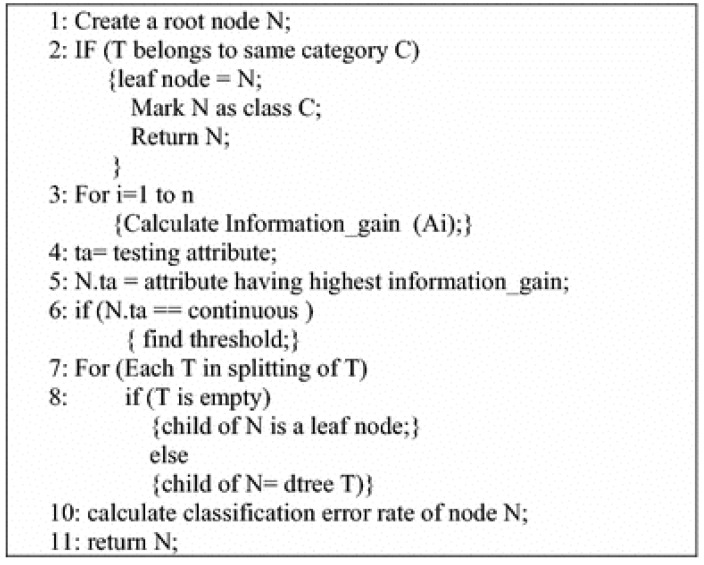
Pseudocode of C4.5 algorithm [26].

**Figure 4 genes-14-00284-f004:**
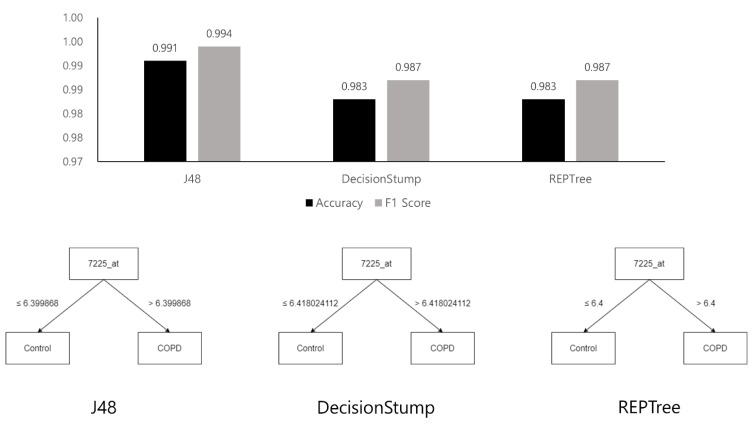
Chart visualizing the performance of the three decision tree models and their structures.

**Figure 5 genes-14-00284-f005:**
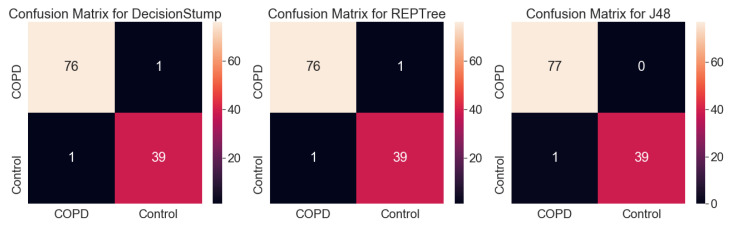
Confusion matrix for results shown in Figure 4. As a result of the experiment using machine learning models, the COPD and control groups of the dataset were classified only according to the expression level of TRPC6, regardless of the values of the other genes, and the accuracy of the best model (J48) was over 99%. This result shows that the classifiers that the decision tree algorithm generated classified COPD patients and controls with simple criteria but high accuracy.

**Figure 6 genes-14-00284-f006:**
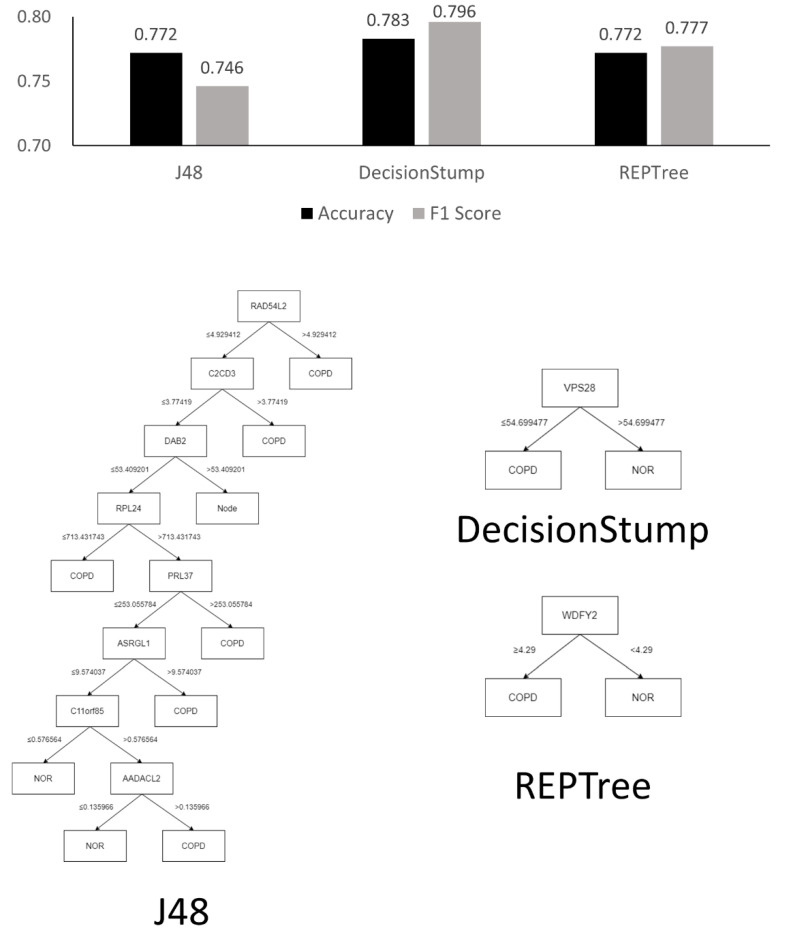
Chart for visualizing the performance of the three decision tree models with the GSE57148 dataset [30] and their structure.

**Figure 7 genes-14-00284-f007:**
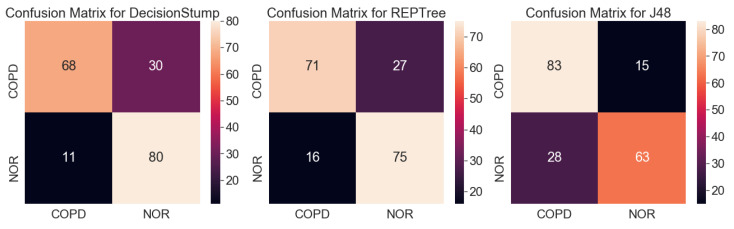
Confusion matrices for results showed in Figure 6.

**Figure 8 genes-14-00284-f008:**
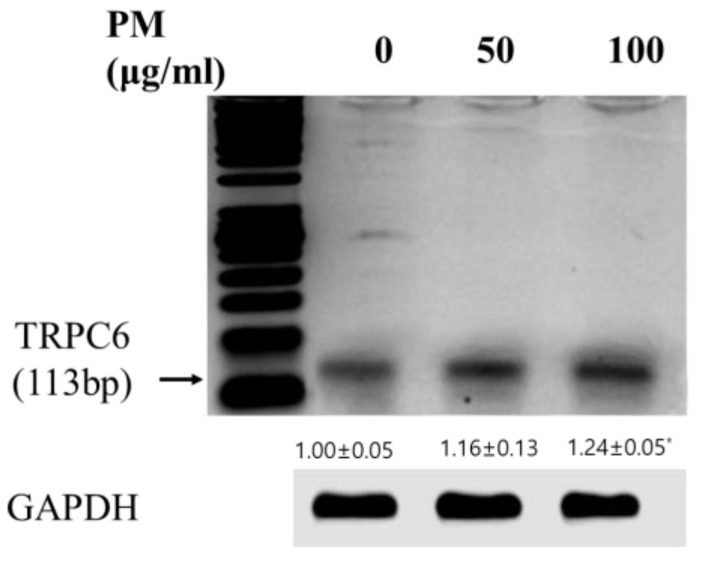
Determination of the TRPC6 mRNA expression in RAW 264.7 cells. Data are expressed as mean ± standard deviation (SD) of 3 independent experiments (*n* ≥ 3). The mRNA band intensities were converted to the numerical data using Image-Studio software (LI COR, Lincoln, NE, USA). One-way ANOVA was used for comparison of group means, followed by Dunnett’s t-test (* *p* < 0.05 vs. untreated control).

**Table 1 genes-14-00284-t001:** Confusion matrix.

Actual	COPD	Control
Prediction		
COPD	True Positive (TP)	False Positive (FP)
Control	False Negative (FN)	True Negative (TN)

**Table 2 genes-14-00284-t002:** List of differentially expressed total RNAs using GEO2R.

No.	ID	Gene	Adjusted *p* Value	*p* Value	logFC
Upregulated genes (top 10)					
1	55282_at	LRRC36	7.54 × 10^−23^	2.45 × 10^−25^	1.377
2	344148_at	NCKAPS	1.87 × 10^−27^	1.71 × 10^−30^	1.251
3	2487_at	FRZB	6.54 × 10^−21^	3.52 × 10^−23^	1.250
4	6092_at	ROBO2	9.24 × 10^−25^	1.64 × 10^−27^	1.159
5	6387_at	CXCL12	4.81 × 10^−19^	4.20 × 10^−21^	1.129
6	653_at	BMP5	6.71 × 10^−20^	4.46 × 10^−22^	1.128
7	2669_at	GEM	3.15 × 10^−19^	2.58 × 10^−21^	1.116
8	7225_at	TRPC6	1.15 × 10^−30^	4.08 × 10^−34^	1.110
9	84251_at	SGIP1	3.26 × 10^−27^	3.31 × 10^−30^	1.062
10	114905_at	C1QTNF7	2.09 × 10^−30^	8.48 × 10^−34^	1.062
Downregulated genes					
1	9332_at	CD163	2.33 × 10^−24^	4.73 × 10^−27^	−2.271
2	6036_at	RNASE2	3.00 × 10^−26^	3.65E × 10-^29^	−1.567
3	29968_at	PSAT1	1.46 × 10^−24^	2.60 × 10^−27^	−1.34
4	4830_at	NME1	1.05 × 10^−22^	3.71 × 10^−25^	−1.318
5	1646_at	AKR1C2	8.76 × 10^−20^	6.22 × 10^−22^	−1.221
6	1510_at	CTSE	3.28 × 10^−18^	3.90 × 10^−20^	−1.208
7	195814_at	SDR16C5	3.13 × 10^−20^	1.90 × 10^−22^	−1.049
8	123_at	PLIN2	5.33 × 10^−24^	1.32 × 10^−26^	−1.039
9	3855_at	KRT7	1.01 × 10^−22^	3.47 × 10^−25^	−1.035
10	6472_at	SHMT2	2.58 × 10^−24^	5.37 × 10^−27^	−1.019

**Table 3 genes-14-00284-t003:** Top five gene ontology (GO) terms and Kyoto Encyclopedia of Genes and Genomes (KEGG) pathways enriched for upregulated DEGs.

Category	Term	Count	%	*p*-Value	Benjamin
GOTERM_BP_DIRECT	Transcription, DNA-templated	14	15.9	8.40 × 10^−02^	1.00 × 10^0^
GOTERM_BP_DIRECT	Transmembrane transport	4	4.5	9.30 × 10^−02^	1.00 × 10^0^
GOTERM_BP_DIRECT	Postembryonic development	3	3.4	4.10 × 10^−02^	1.00 × 10^0^
GOTERM_BP_DIRECT	Ossification	3	3.4	4.90 × 10^−02^	1.00 × 10^0^
GOTERM_BP_DIRECT	Covalent chromatin modification	3	3.4	8.90 × 10^−02^	1.00 × 10^0^
GOTERM_CC_DIRECT	Plasma membrane	29	33	3.50 × 10^−03^	2.60 × 10^−01^
GOTERM_CC_DIRECT	Intracellular	11	12.5	4.90 × 10^−02^	1.00 × 10^0^
GOTERM_CC_DIRECT	Nuclear chromatin	6	6.8	1.30 × 10^−03^	1.90 × 10^−01^
GOTERM_CC_DIRECT	Sarcolemma	3	3.4	4.90 × 10^−02^	1.00 × 10^0^
GOTERM_CC_DIRECT	Receptor complex	3	3.4	9.80 × 10^−02^	1.00 × 10^0^
GOTERM_MF_DIRECT	DNA binding	14	15.9	3.20 × 10^−02^	1.00 × 10^0^
GOTERM_MF_DIRECT	Transcription factor activity, sequence-specific DNA binding	9	10.2	6.30 × 10^−02^	1.00 × 10^0^
GOTERM_MF_DIRECT	Calcium ion binding	8	9.1	4.00 × 10^−02^	1.00 × 10^0^
GOTERM_MF_DIRECT	Chromatin binding	7	8	7.80 × 10^−03^	9.90 × 10^−01^
GOTERM_MF_DIRECT	Integrin binding	4	4.5	1.10 × 10^−02^	9.90 × 10^−01^
KEGG_PATHWAY	Axon guidance	5	5.7	2.10 × 10^−03^	2.30 × 10^−01^
KEGG_PATHWAY	Serotonergic synapse	3	3.4	8.40 × 10^−02^	1.00 × 10^0^
KEGG_PATHWAY	Pathways in cancer	5	5.7	8.90 × 10^−02^	1.00 × 10^0^

**Table 4 genes-14-00284-t004:** Top five gene ontology (GO) terms and Kyoto Encyclopedia of Genes and Genomes (KEGG) pathways enriched for downregulated DEGs.

Category	Term	Count	%	*p*-Value	Benjamin
GOTERM_BP_DIRECT	Oxidation–reduction process	15	10.3	3.20 × 10^−04^	6.70 × 10^−02^
GOTERM_BP_DIRECT	tRNA aminoacylation for protein translation	8	5.5	2.90 × 10^−08^	2.30 × 10^−05^
GOTERM_BP_DIRECT	Cell–cell adhesion	8	5.5	6.60 × 10^−03^	7.50 × 10^−01^
GOTERM_BP_DIRECT	IRE1-mediated unfolded protein response	7	4.8	8.00 × 10^−06^	3.20 × 10^−03^
GOTERM_BP_DIRECT	Response to nutrient	6	4.1	3.30 × 10^−04^	6.70 × 10^−02^
GOTERM_CC_DIRECT	Extracellular exosome	51	34.9	2.00 × 10^−09^	4.40 × 10^−07^
GOTERM_CC_DIRECT	Cytoplasm	53	36.3	1.60 × 10^−02^	2.40 × 10^−01^
GOTERM_CC_DIRECT	Cytosol	50	34.2	1.20 × 10^−06^	1.30 × 10^−04^
GOTERM_CC_DIRECT	Membrane	32	21.9	4.80 × 10^−04^	1.70 × 10^−02^
GOTERM_CC_DIRECT	Mitochondrion	27	18.5	7.70 × 10^−06^	5.50 × 10^−04^
GOTERM_MF_DIRECT	NADP binding	6	4.1	7.60 × 10^−06^	2.50 × 10^−03^
GOTERM_MF_DIRECT	Poly(A) RNA binding	21	14.4	5.30 × 10^−04^	7.60 × 10^−02^
GOTERM_MF_DIRECT	ATP binding	21	14.4	1.30 × 10^−02^	5.50 × 10^−01^
GOTERM_MF_DIRECT	Cadherin binding involved in cell–cell adhesion	8	5.5	8.10 × 10^−03^	3.90 × 10^−01^
GOTERM_MF_DIRECT	Protein kinase binding	7	4.8	7.70 × 10^−02^	1.00 × 10^0^
KEGG_PATHWAY	Biosynthesis of antibiotics	16	11	1.20 × 10^−08^	1.50 × 10^−06^
KEGG_PATHWAY	Metabolic pathways	33	22.6	1.50 × 10^−06^	6.10 × 10^−05^
KEGG_PATHWAY	Biosynthesis of amino acids	9	6.2	1.40 × 10^−06^	6.10 × 10^−05^
KEGG_PATHWAY	Carbon metabolism	9	6.2	4.10 × 10^−05^	1.20 × 10^−03^
KEGG_PATHWAY	Aminoacyl–tRNA biosynthesis	7	4.8	9.90 × 10^−05^	2.40 × 10^−03^

## Data Availability

Not applicable.

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
