# Peer review of "Identification of TRPC6 as a Novel Diagnostic Biomarker of PM-Induced Chronic Obstructive Pulmonary Disease Using Machine Learning Models"

_genes, 2023, doi:10.3390/genes14020284_

Round 1
Reviewer 1 Report
In this manuscript, Park et. al have attempted to decipher the role of TRPC6 as a novel diagnostic biomarker of 2 pm-induced chronic obstructive pulmonary disease using machine learning models. Collectively, the study is well conducted. However, I would ask the authors to address the following comments to further improve the overall quality of the manuscript.
1. Line 21 of Abstract: “The 21 GEO2R analysis revealed that TRPC6 was the sixth most highly expressed gene in patients with 22 COPD”. Authors should explain the reason to select sixth most expressed gene and not the other five, clarify in abstract.
2. Line 66 Introduction: Authors said “we aimed to identify novel potential diagnostic biomarkers of COPD” but they have specifically mentioned only one biomarker in title of the manuscript and also in the abstract. Authors can mention some other biomarkers as well along with the mentioned one to justify their aim of this study.
3. Figure 1. A brief schema of decision tree. Instead of giving information on decision tree only, example of how decision tree is used for this problem should be given. Authors may customized the decision tree based on their problem, which will make it more useful to the readers.
4. Conclusions can be expanded describing the computational and experimental studies.
5. Authors can give a workflow figure in methodology describing step by step methodology of the manuscript work.
Author Response
Dear Editor and Reviewer
We greatly appreciate the efforts of the Editorial Board and reviewers for giving us the valuable comments on our manuscript for publication in Genes. Here, we have enclosed a revised version of manuscript in response to the reviewer comments. Included below is a point-by-point description of our responses to the Editor’s and reviewer’s comments. The revision made in the manuscript was written in blue. We cordially hope that you and the reviewers find this revised manuscript acceptable for publication in Genes.
Sincerely,
Hye-Jin Park

Reviewer 2 Report
Review Report
There are plenty of previous studies indicating functional role of TRPC6 channels on human lung macrophage physiology, and potential association with pulmonary fibrosis, COPD etc. So, there is nothing new about role of this gene in COPD pathogenesis.
In present report, authors claim this gene as a novel COPD biomarker based on their results of the analysis of GEO dataset and a machine learning model. Authors also use a cell culture system where they observe the upregulation of TRPC6 in PM-stimulated RAW264.7 cells (that mimics COPD conditions) vs. non-stimulated cells. However, there are some Major concerns that are raised based on their findings as below –
1. Authors received the DEGs based on |log (fold change; FC) | >1 and t- tests with p<0.05. This seems to be relaxed cutoff and may involve potential false positive DEGs. Did authors perform any multiple testing correction? What are the top 10 DEGs when authors apply a stringent Bonferroni correction?
2. It is recommended to analyze the same GEO dataset using two or more independent tools such as – Edge R, DEseq2 etc. to validate the robustness of ranking of top10 genes reported in present manuscript.
3. It is recommended to share the raw Ct values for test and reference genes of the qPCR experiments with melt curve analysis in the supplementary files to visualize the accuracy of the experiments performed, which in turn shape the conclusions of the manuscript.
4. How many replicates were used for quantitative polymerase chain reaction? Details should be incorporated in the manuscript.
5. Authors reported an upregulation in TRPC6 gene expression in PM-stimulated RAW264.7 cell vs. non stimulated cells. However, it raises the interest that whether the upregulation of TRPC6 was only at the RNA level or it also persisted at protein levels. A relative comparison of TRPC6 protein levels in PM-stimulated RAW264.7 cells vs. non stimulated cells (Can be addressed using the western blot analysis) can help to address this concern and is strongly recommended.
6. Authors looked at gene expression of TRPC6 using quantitative reverse transcription polymerase chain reaction. Did authors also perform any immuno-histochemistry experiments to visualize changes in TRPC6 transcripts localization, macrophage cells proliferation in PM-stimulated RAW264.7 vs. non stimulated cells?
7. Current sample size (117 samples) seems to very low when authors really want to identify a biomarker for COPD. What is the power of the present study for claiming the gene to be a “biomarker”.
8. The machine learning model ranks TRPC6 as the most highly expressed gene among top 10 upregulated genes in patients with COPD compared to the control. However, GEO2R analysis indicates RTKN2 as the most abundant whereas TRPC6 as the 8th highest abundant gene among the top 10 upregulated genes in patients with COPD compared to the control. It is not clear why do authors recommend the machine learning gene candidate as the biomarker preferentially in the manuscript. It is recommended to compute validation rate of gene hits obtained using differential expression analysis vs. machine learning model.
9. Based on machine learning decision tree, only one gene (TRPC6) was used for the classification, which needs a strong explanation as to why only one gene was used for classification and how can we classify the gene as a biomarker based on just this analysis which the gene expression datasets with fair sample sizes never ranked among the top. What is the major difference in gene ranking among two methods is still a question mark in the manuscript and needs to be clearly addressed as limitation of the study.
10. What is the sensitivity and specificity of the machine learning model used? Why were the current model chosen and how it is better than other existing models?
11. Since genes work synergistically, it will be interesting to investigate differential co-expressed genes network in COPD vs. Normal samples instead of just highlighting single gene as a major biomarker for disease diagnosis. The co-expressed gene sets can be robust biomarkers instead of a single gene.
12. Can authors test their machine learning model in another independent COPD vs. normal GEO dataset and do they identify the similar or different gene hits will be something interesting to explore given the additional dataset availability {GSE57148}.
13. How were the PM particles prepared/derived? What is their size? How long the cells were exposed to PM particles treatment?
14. In figure 6 - A quantitative figure with unpaired t-test is recommended to see significant comparison among groups (demonstrated with error bars and p-values). Also, the size of the product bands is not shown in the figure. It is desired to add the DNA standard size also in the figure.
Minor comments –
1. It is not clear which normalization method was used by the GEO2R tool for normalization of RNAseq data. The details should be added in the manuscript
2. What are a and b in figure 6 legends are not clear.
Author Response

(The authors gave the same response as above.)

Round 2
Reviewer 2 Report
I appreciate to see that authors have significantly improved the manuscript with better-quality analyses and revised experiments in short period of time which is a great job. However, I still have some minor comments that needs authors attention:
1.To address the comment1, authors chose to perform Benjamin and Hochberg False Discovery Rate and Bonferroni corrections in the filtering menu of GEO2R. However, there seems to be difference in the list of top 10 genes that has been shown in the main table 1 (manuscript) using the relaxed cutoff of log (fold change; FC) | >1 and t- tests with p<0.05. vs. the table1 (review response letter) using multiple correction tests. Interestingly, TRPC6 (7225_at) still pops up in the new analysis, but it seems to be downregulated based on logFC in contrast to main table1 in the manuscript where it is upregulated. Why do we observe opposite effects? Although we do observe that there was no difference in the rank of top 10 genes in Benjamin & Hochberg vs. Bonferroni corrected results, but the overall ranking of top10 genes indeed changes when we use these multiple testing corrections.
3. I do not agree that DESeq2 package is not easy to access to biologists. It is easily accessible, a user-friendly and robust tool and should be considered for the most of differential analyses in the future.
4. It is not clear why the authors chose to perform quantile normalization that has potential to reduce the true biological variability between sample groups, while there are other available better methods of data normalization to address the data normalization issues. It is recommended to avoid this normalization for future analyses without a strong reason.
5. What do the values/numbers represent that have been used for calculating group means and Dunnett’s t-test in figure 8? Since authors mention they have used reverse transcriptase PCR and not a quantitative real-time PCR, how do they get these values? Details should be incorporated in the figure legends.
Author Response

(The authors gave the same response as above.)
